# EVALUATING REPRESENTATIONAL SIMILARITY MEASURES FROM THE LENS OF FUNCTIONAL CORRESPONDENCE

## ABSTRACT

Neuroscience and artificial intelligence (AI) both face the challenge of interpreting high-dimensional neural data, where the comparative analysis of such data is crucial for revealing shared mechanisms and differences between these complex systems. Despite the widespread use of representational comparisons and the abundance classes of comparison methods, a critical question remains: which metrics are most suitable for these comparisons? While some studies evaluate metrics based on their ability to differentiate models of different origins or constructions (e.g., various architectures), another approach is to assess how well they distinguish models that exhibit distinct behaviors. To investigate this, we examine the degree of alignment between various representational similarity measures and behavioral outcomes, employing group statistics and a comprehensive suite of behavioral metrics for comparison. In our evaluation of eight commonly used representational similarity metrics in the visual domain—spanning alignment-based, Canonical Correlation Analysis (CCA)-based, inner product kernel-based, and nearest-neighbor methods—we found that metrics like linear Centered Kernel Alignment (CKA) and Procrustes distance, which emphasize the overall geometric structure or shape of representations, excelled in differentiating trained from untrained models and aligning with behavioral measures, whereas metrics such as linear predictivity, commonly used in neuroscience, demonstrated only moderate alignment with behavior. These insights are crucial for selecting metrics that emphasize behaviorally meaningful comparisons in NeuroAI research.

## 1 INTRODUCTION

Both neuroscience and artificial intelligence (AI) confront the challenge of high-dimensional neural data, whether from neurobiological firing rates, voxel responses, or hidden layer activations in artificial networks. Comparing such high-dimensional neural data is critical for both fields, as it facilitates understanding of complex systems by revealing their underlying similarities and differences.

In neuroscience, one of the main goals is to uncover how neural activity drives behavior and to understand neural computations at an algorithmic level. Comparisons across species and between brain and model representations, particularly those of deep neural networks, have been instrumental in advancing this understanding (Yamins et al. (2014); Eickenberg et al. (2017); Güçlü & Van Gerven (2015); Cichy et al. (2016); Khaligh-Razavi & Kriegeskorte (2014); Schrimpf et al. (2018; 2020); Storrs et al. (2021); Kriegeskorte et al. (2008)). A growing interest lies in systematically altering model parameters—such as architecture, learning objectives, and training data—and comparing the resulting internal representations with neural data (Yamins & DiCarlo (2016); Doerig et al. (2023); Schrimpf et al. (2018; 2020)).

Similarly, in AI, researchers are increasingly focused on reverse-engineering neural networks by tweaking architectural components, training objectives, and data inputs to examine how these modifications impact the resulting representations. However, studying neural networks in isolation can be limiting, as interactions between the learning algorithms and structured data shape these systems in ways we do not yet fully understand. Comparative analysis of model representations offers a powerful tool to probe these networks more deeply. This endeavor is rooted in the universality

hypothesis that similar phenomena can arise across different networks. Indeed, a large number of studies have provided empirical evidence licensing these universal theories (Huh et al. (2024); Kornblith et al. (2019); Bansal et al. (2021); Li et al. (2015); Roeder et al. (2021); Lenc & Vedaldi (2015)) but the extent to which diverse neural networks converge to similar representations is not well understood.

Given the growing interest in comparative analyses across neuroscience and AI, a key question arises: what are the best tools for conducting such analyses? Over the past decade, a wide variety of approaches have emerged for quantifying the representational similarity across artificial and biological neural representations (Sucholutsky et al. (2023); Klabunde et al. (2023); Williams et al. (2021)). Most of these approaches can be classified as belonging to one of four categories: representational similarity based measures, alignment-based measures, nearest-neighbor based measures and canonical correlation analysis-based measures (Klabunde et al. (2023)). With the wide range of available approaches for representational comparisons, researchers are tasked with selecting a suitable metric. The choice of a specific metric implicitly prioritizes certain properties of the system, as different approaches emphasize distinct invariances and are sensitive to varying aspects of the representations. This complexity ties into broader issues in the concept and assessment of similarity, which, as emphasized in psychology, is highly context-dependent (Tversky (1977)).

What, then, are the key desiderata for network comparison metrics? Networks may exhibit similarities in some dimensions and differences in others, but the critical question is whether these differences are functionally relevant or merely reflect differences in origin or construction. This consideration leads to a central criterion for effective metrics: behavioral differences should correspond to differences in internal representational similarity (Cao (2022)). However, identifying which measures reliably capture behaviorally meaningful differences remains an open question.

Our study aims to address the above challenge. Here, we make the following key contributions:

- We conduct an extensive analysis of common representational comparison measures (including alignment-based, representational similarity matrix-based, CCA-based, and nearest-neighbor-based methods) and show that these measures differ in their capacity to distinguish between models. While some measures excel at distinguishing between models from different architectural families, others are better at separating trained from untrained models.
- To assess which of these distinctions reflects differences in model behaviors, we perform complementary behavioral comparisons using a comprehensive set of behavioral metrics (both hard and soft prediction-based). We find that behavioral metrics are generally more consistent with each other than representational similarity measures.
- Finally, we cross-compare representational and behavioral similarity measures, revealing that linear CKA and Procrustes distance align most closely with behavioral evaluations, whereas metrics like linear predictivity, widely used in neuroscience, show only modest alignment. This finding offers important guidance for metric selection in neuroAI, where the functional relevance of representational comparisons is paramount.

**Related Work** Although few studies directly compare representational similarity measures based on their discriminative power, most efforts in this area focus on identifying metrics that distinguish between models by their construction. These efforts typically involve assessing measures based on their ability to match corresponding layers across models with varying seeds (Kornblith et al., 2019) or identical architectures with different initializations (Han et al., 2023; Rahamim & Belinkov, 2024). The closest to our work are studies by Ding et al. (Ding et al., 2021) and Cloos et al. (Cloos et al., 2024). Cloos et al. (Cloos et al., 2024) optimized synthetic datasets to resemble brain activity under various measures, demonstrating that metrics like linear predictivity and CKA can yield high scores even when task-relevant variables are not encoded. Ding et al. ((Ding et al., 2021)) examined the sensitivity of representational similarity measures—CCA, CKA, and Procrustes—in BERT models (NLP) and ResNet models (CIFAR-10) to factors that either preserve functional behavior (e.g., random seed variations) or alter it (e.g., principal component deletion). However, these studies examine a limited set of similarity measures and primarily assess functional similarity based on task performance alone, without evaluating the finer-grained alignment of predictions across models.

## 1.1 METRICS FOR REPRESENTATIONAL COMPARISONS

**Notations and Definitions** Let $S$ be a set of $M$ fixed input stimuli. Define the kernel functions [1] $f : S \to \mathbb{R}^{N_X}$ and $g : S \to \mathbb{R}^{N_Y}$, where $N_X$ and $N_Y$ are the output unit sizes of the first and second encoders, respectively. Here, $f(s_i)$ and $g(s_i)$ map each stimulus $s_i \in S$ to vectors in $\mathbb{R}^{N_X}$ and $\mathbb{R}^{N_Y}$.

Let $X \in \mathbb{R}^{M \times N_X}$ and $Y \in \mathbb{R}^{M \times N_Y}$ be the representation matrices. For each input stimulus $s_i$, denote the $i$th row of $X$ as $\phi_i = f(s_i)$ and of $Y$ as $\psi_i = g(s_i)$, each being the activation in response to the $i$th stimulus.

**Representational Similarity Analysis (RSA)** (Kriegeskorte et al., 2008) A method that quantifies the distance between $M \times M$ Representational Dissimilarity Matrices (RDMs) of two models in response to a common set of $M$ stimuli.

$$\mathrm{RSA}(X, Y) = \tau(\mathbf{J}_M - X^T X, \mathbf{J}_M - Y^T Y)$$

with $J_M$ denoting the $M \times M$ all-ones matrix, the representational dissimilarity matrices (RDMs) for $X$ and $Y$ are $J_M - X^T X$ and $J_M - Y^T Y$, respectively. $X^T X$ and $Y^T Y$ in $\mathbb{R}^{M \times M}$ represent the self-correlations of $X$ and $Y$, with each matrix entry $i, j$ quantifying the correlation between activations for the $i^{th}$ and $j^{th}$ stimuli. The Kendall rank correlation coefficient $\tau(\cdot)$ quantifies the similarity between these RDMs.

**Canonical Correlation Analysis (CCA)** (Hotelling, 1992) A popular linear-invariant similarity measure quantifying the multivariate similarity between two sets of representations $X$ and $Y$ under a shared set of $M$ stimuli by identifying the bases in the unit space of matrix $X$ and $Y$ such that when the two matrices are projected on to these bases, their correlation is maximized.

Here, the $i^{th}$ canonical correlation coefficient $\rho_i$ (associated with the $i^{th}$ optimized canonical weights $w_x^i \in \mathbb{R}^{N_X}$ and $w_y^i \in \mathbb{R}^{N_Y}$) is being calculated by:

$$\rho_i = \max_{w_x^i, w_y^i} \mathrm{corr}(X w_x^i, Y w_y^i)$$

$$\text{subject to } \forall j < i, \quad X w_x^i \perp X w_x^j \quad \text{and} \quad Y w_y^i \perp Y w_y^j,$$

with the transformed matrices $X w_x^i$ and $Y w_y^i$ being called canonical variables.

To obtain a measure of similarity between neural network representations, the mean CCA correlation coefficient $\bar{\rho}$ over the first $N'$ components is reported, with $N' = \min(N_X, N_Y)$. Here,

$$\bar{\rho} = \frac{\sum_{i=1}^{N'} \rho_i}{N'} = \frac{\left\| Q_Y^T Q_X \right\|_*}{N'},$$

where $\| \cdot \|_*$ denotes the nuclear norm. Here, $Q_X = X(X^T X)^{-1/2}$ and $Q_Y = Y(Y^T Y)^{-1/2}$ represent any orthonormal bases for the columns of $X$ and $Y$.

**Linear Centered Kernel Alignment (CKA)** (Kornblith et al., 2019; Gretton et al., 2005)

A representation-level comparison that measures how (in)dependent the two models' RDMs are under a shared set of $M$ stimuli. This measure possesses a weaker invariance assumption than CCA, being invariant only to orthogonal transformations, rather than all classes of invertible linear transformations, which implies the preservation of scalar products and Euclidean distances between pairs of stimuli.

$$\mathrm{CKA}(K, L) = \frac{\mathrm{HSIC}(K, L)}{\sqrt{\mathrm{HSIC}(K, K)\mathrm{HSIC}(L, L)}}$$

with $K$ and $L$ be kernel matrices where $K_{ij} = \kappa(\phi_i, \phi_j)$ and $L_{ij} = \kappa(\psi_i, \psi_j)$. These matrices represent the inner products of vectorized features $\phi$ and $\psi$ from two different models, respectively, computed using the kernel function function $\kappa$. In the linear case, $\kappa$ is the inner product, implying $K = XX^T, L = YY^T$. The Hilbert-Schmidt Independence Criterion $\mathrm{HSIC}(\cdot)$ evaluates the cross-covariance of the models' internal embedding spaces, focusing on the similarity of stimulus pairs.

---

[1]The term "encoder/kernel function: refers to the function that represents the mapping from an input to the output of a specific layer's activation in a neural network

**Mutual k-nearest neighbors** (Huh et al., 2024) A local-biased representation-level measure that quantifies the similarity between the representations of two models by assessing the average overlap of their nearest neighbor sets for corresponding features.

$$\text{MNN}(\phi_i, \psi_i) = \frac{1}{k}|S(\phi_i) \cap S(\psi_i)|$$

where $\phi_i = f(s_i)$ and $\psi_i = g(s_i)$ are features derived from model representations $f$ and $g$ given the shared stimulus $s_i$. $S(\phi_i)$ and $S(\psi_i)$ are the set of indices of the $k$-nearest neighbors of $\phi_i$ and $\psi_i$ in their respective feature spaces and $|\cdot|$ is the size of the intersection.

**Linear predictivity** An asymmetric measure of alignment between the representations of two systems, obtained using ridge regression. The numerical score is calculated by summing Pearson's correlations between each pair of predicted and actual activations in the held-out set. For reporting, we provide symmetrized scores by averaging the correlation coefficients from both fitting directions.

**Procrustes distance** (Ding et al., 2021; Williams et al., 2021) A rotational-invariant shape align-ment distance between $X$ and $Y$'s representations after removing the components of uniform scaling and translation and applying an optimized mapping, where the mappings from one representation matrix to another is constrained to rotations and reflection. Here, the Procrustes distance is given by:

$$d(X, Y) = \min_{T \in O(n)} \|\phi(X) - \phi(Y)T\|_F$$

where $\phi(\cdot)$ is the function that whitens the covariance of the matrix X and Y, i.e. the columns sum to zero and $\|\phi(X)\|_F, \|\phi(X)\|_F = 1$. $O(n)$ is the orthogonal group.

The similarity scores reported are obtained by $1 - d(X, Y)$, such that the comparison with a repre-sentation itself yields a score of 1, and lower distance yields a higher score.

**Semi-matching score** (Li et al., 2015; Khosla et al., 2024) An asymmetric correlation-based measure obtained using the average correlation after matching every neuron in $X$ to its most similar partner in $Y$. The scores reported are the average from both fitting directions.

$$s_{\text{semi}}(X, Y) = \frac{1}{N_x} \sum_{i=1}^{N_x} \max_{j \in \{1, \ldots, N_y\}} x_i^\top y_j$$

**Soft-matching distance** (Khosla & Williams, 2024) A generalization of permutation dis-tance (Williams et al., 2021) to representations with different number of neurons. It measures alignment by relaxing the set of permutations to "soft permutations". Specifically, consider a non-negative matrix $\in^{N_x \times N_y}$ whose rows each sum to $1/N_x$ and whose columns each sum to $1/N_y$. The set of all such matrices defines a *transportation polytope* (De Loera & Kim, 2013), denoted as $\text{T}(N_x, N_y)$. Optimizing over this set of rectangular matrices results in a "soft matching" or "soft permutation" of neuron labels in the sense that every row and column of P may have more than one non-zero element.

$$d_{\text{T}}(X, Y) = \sqrt{\min_{P \in \text{T}(N_X, N_Y)} \sum_{i,j} P_{ij}\|x_i - y_j\|^2}$$

## 1.2 DOWNSTREAM BEHAVIORAL MEASURES

For classification tasks, we incorporate various downstream measurements at different levels of granularity to assess behavioral consistency across systems. For a given pair of neural networks, their activations over a shared set of stimuli are extracted. A linear readout based on a fully connected layer is trained over a training set of activations, where the resulting behavioral classification decisions determined by the linear readouts on a held-out testing set are exploited in the following ways as a comparison between the neural networks:

Figure 1: Framework for evaluating representational similarity metrics based on their functional correspondence. We conduct pairwise comparisons of the representational similarities and behavioral outputs of 19 vision models, utilizing 9 widely-used representational similarity measures and 10 behavioral metrics across 17 distinct behavioral datasets.

**Raw Softmax alignments** emphasize the consistency of numerical class-level activation strength patterns. Compares two models' representations by their linear-readout's softmax layer activation, which is a class-dimensional vector reflecting the model's judgement of the probabilities assigned to each label for a given input, with scores calculated by summing the Pearson correlation coefficient between these softmax vectors over the testing set.

**Classification Confusion Matrix alignments** emphasize the consistency of discrete inter-class (mis) classification patterns. A similarity score is obtained by comparing the two models' confusion matrices in the following ways:

1 **Pearson Correlation Coefficient** between the flattened confusion matrices given by two models, each being a vector of dimension $C^2$ over $C$ classes.

2 **Jensen-Shannon (JS) Distance** (Lin, 1991) introduced as a behavioral alignment measure by Tuli et al. (2021) is functionally similar to a symmetrized and smoother version of the Kullback-Leibler (KL) divergence. For class-wise JS distance, let $\hat{p} = \langle p_1, p_2, \ldots, p_C \rangle$ and $\hat{q} = \langle q_1, q_2, \ldots, q_C \rangle$ be error probability vectors over C classes, with

$$p_i = \frac{e_i}{\sum_{i=1}^{C} e_i}, \forall i \in \{1, 2, ..., C\}$$

where $e_i$ represents error counts per class. The JS divergence is defined as:

$$JSD(p, q) = \sqrt{\frac{D(p||m) + D(q||m)}{2}},$$

$$\text{with } D(p||m) = \sum_{i=1}^{C} p_i \log\left(\frac{p_i}{m_i}\right) \text{ and } m_i = \frac{p_i + q_i}{2}$$

A finer inter-class dissimilarity measure derived from the complete misclassification patterns shown in the non-diagonal elements of the confusion matrix results in two $C * (C - 1)$ dimensional flattened vectors $\hat{p}$ and $\hat{q}$, where each component is proportional to the counts of misclassifications from class $i$ to class $j$, is calculated as

$$\frac{e_{ij}}{\sum_{i=1}^{C} \sum_{j=1, j \neq i}^{C} e_{ij}}, \quad \forall i, j \in \{1, 2, \ldots, C\}$$

.

The resulting distances from both method range from [0, 1], where we simply report a similarity measure given by $1 - JSD(p, q)$.

**Classification Binary Correctness alignments** emphasize consistency in per-stimulus prediction correctness. The error patterns for each model are encoded as vectors of binary values, where each entry corresponds to the correctness of a stimulus's prediction. We incorporate the following measures to compare alignment between the binary vectors:

1. **Pearson Correlation Coefficient** between the two binary vectors of dimension $M$ over $M$ shared testing stimuli, reflecting the prediction correctness of two models (1 = correct, 0 = incorrect).

2. **Cohen's $\kappa$ Score** Consider two systems tested independently on identical trials, each correctly classifying with a probability $p_{correct}$, leading to i.i.d. samples from a binomial distribution.

$$\kappa_{xy} = \frac{c_{obs,xy} - c_{exp,xy}}{1 - c_{exp,xy}},$$

with $c_{exp,xy} = p_i p_j + (1 - p_i)(1 - p_j)$, $c_{obs,xy} = $ # of agreements$/M$

where $c_{exp,xy}$ represents the expected probability of agreement between model $x$ and $y$, calculated from the accuracies $p_x$ and $p_y$ of two independent binomial observers, and $c_{obs,xy}$ denotes the observed probability of agreement. Cohen's $\kappa$ assesses the consistency of error overlap, providing a measure of classification agreement without distinguishing error types.

3. **Jaccard Similarity Coefficient** is defined as:

$$J(x, y) = \frac{\sum_{i=1}^{n} x_i y_i}{\sum_{i=1}^{n} (x_i + y_i - x_i y_i)}$$

where each $x_i, y_i \in \{0, 1\}$ represents the correctness (1) or incorrectness (0) of the $i$th sample prediction from the two models, respectively. The numerator "|Intersections|" counts samples where both models predict correctly, normalized by "|Unions|", which counts samples where either model predicts correctly.

4. **Hamming Distance** counts the number of discrepancies in the correctness of predictions:

$$d(x, y) = |\{i : x_i \neq y_i, i = 1, \ldots, n\}|.$$

5. **Agreement Score** is the normalized difference between counts of agreement and disagreement in the prediction correctness made by the two models:

$$s(x, y) = \frac{(n_{11} + n_{00}) - (n_{10} + n_{01})}{n_{11} + n_{00} + n_{10} + n_{01}}$$

with $n_{ij}$, where $i, j \in 0, 1$, counts predictions where model $x$ predicts $i$ (correct/incorrect) and model $y$ predicts $j$ over shared stimuli.

## 1.3 DOWNSTREAM BEHAVIORAL DATASETS

We analyze the behavior of all models across a series of downstream tasks, including in-distribution and several out-of-distribution image types, such as silhouettes, stylized images, and natural images distorted by various noise types (see Appendix A.1 for details). In total, these comparisons span 17 behavioral datasets.

## 1.4 SELECTION OF NEURAL NETWORK ARCHITECTURES AND LAYERS

We incorporated a comprehensive list of popular deep learning models pretrained over the 1000-class classification tasks over the ImageNet-1k dataset (Deng et al., 2009). The selection spans a diverse set of architectures, including conventional convolutional neural networks (CNNs) and transformers. These models were trained using various objective functions, both supervised and self-supervised. Specifically, our lineup includes AlexNet (Krizhevsky et al., 2012), ResNet (He et al., 2015), VGG16 (Simonyan & Zisserman, 2015), Inception (Szegedy et al., 2014), ResNeXt (Xie et al., 2017), MoCo (He et al., 2020), ResNet Robust (Engstrom et al., 2019), and several variants of Vision Transformers (ViTs) (Dosovitskiy et al., 2020) such as Vit-b16 and ViT-ResNet (vit on ResNet architecture),

and Swin transformer (Liu et al., 2021). For representational analysis, we mainly focused on the penultimate layer of each model, where we averaged the outputs across channels or patches, as applicable per architecture. For transformer models, we've included outputs from the final GELU activation layers in addition to their penultimate layer.

We included randomized versions of AlexNet, ResNet, ViT, and Swin to study their behavior under random initialization before training.

## 2 RESULTS

### 2.1 DIFFERENT REPRESENTATIONAL SIMILARITY MEASURES HAVE DISTINCT CAPACITIES FOR MODEL SEPARATION

To characterize how different representational similarity measures discriminate models, we first visualize the model-by-model similarity matrices for each measure. We observed that while some measures like the soft-matching distance were effective at differentiating architectural families (Fig. 2, right), others like the Procrustes distance were more sensitive to the effects of training (Fig. 2, left), clearly separating trained from untrained models. Other measures, like linear predictivity, which allow greater flexibility in aligning the two representations, showed limited ability in distinguishing between models trained with different architectures or trained from untrained models (see Appendix A.4 for additional similarity matrices). To quantify these distinctions, we computed $d'$ scores (Appendix A.2) to assess each measure's ability to differentiate two categories of models: (a) those from different architectural families, and (b) those with varying levels of training (trained vs. untrained). Significant differences in $d'$ scores emerged across measures (Fig. 3). For instance, Procrustes achieved $d'$ scores with a mean of 3.70 when separating trained from untrained models across all datasets, while commonly used measures like CCA and linear predictivity produced much lower scores with means of 0.53 and 0.87, respectively. Similarly, some measures were better at discriminating architectural differences, with the soft-matching distance demonstrating the highest discriminability (mean of $d'$ scores = 1.6). Previous studies have also demonstrated that different measures vary in their effectiveness at establishing layer-wise correspondence across networks with the same architecture (Kornblith et al., 2019; Thobani et al.). Considering these differences in how measures distinguish between models, a key question emerges: Which distinctions should we prioritize?

### 2.2 BEHAVIORAL METRICS PRIMARILY REFLECT LEARNING DIFFERENCES OVER ARCHITECTURAL VARIATIONS

To address the question of which separation should be prioritized, we return to our central premise: measures that emphasize functional distinctions should be favored. Therefore, we next evaluated how different behavioral measures (as previously described) distinguish between models. Our results

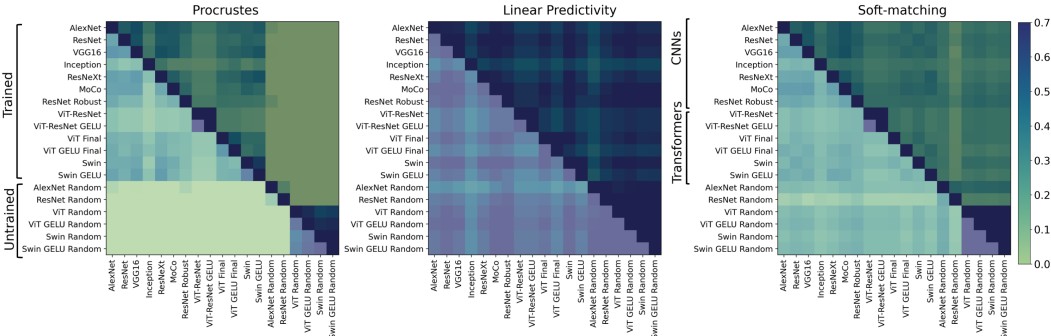

Figure 2: Model-by-model similarity matrices from different measures on the Cue Conflict task. **Left**: The Procrustes measure clearly distinguishes between trained and untrained models. **Middle**: Linear Predictivity reveals no noticeable separation between trained and untrained models or across different architectures. **Right**: Soft-matching more effectively differentiates between architectural families (CNN vs. transformers) compared to other representational metrics.

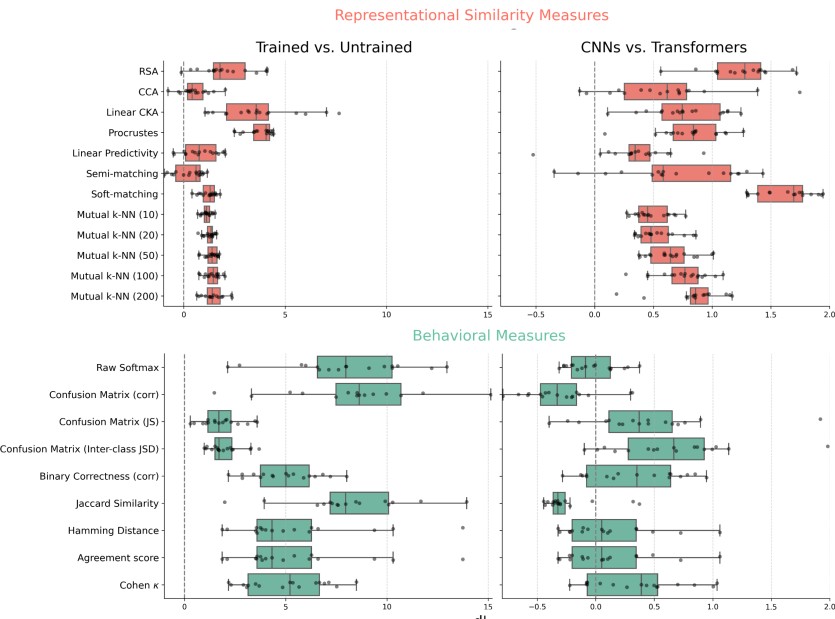

Figure 3: Discriminative ability (d' scores) of (top) representational and (bottom) behavioral similarity measures in distinguishing between trained vs. untrained models (left) and architectures (right).

show that behavioral metrics effectively and consistently separate trained from untrained networks, with even the weakest metric (Confusion Matrix (JSD)) achieving a mean $d'$ of 1.82. However, most behavioral measures struggle to differentiate between architectural families (e.g., CNNs vs. Transformers), with the best-performing metric (Confusion Matrix (Inter-class JSD)) achieving an average $d'$ of 0.65 across all behavioral datasets (see Appendix A.5 for all similarity matrices). This suggests that differences in these architectural motifs have minimal impact on model behavior.

### 2.3 BEHAVIORAL METRICS SHOW GREATER CONSISTENCY THAN NEURAL REPRESENTATIONAL SIMILARITY MEASURES

We next examined the consistency across different representational similarity measures and across different behavioral measures by computing correlations between the model-by-model similarity matrices generated by each measure. As shown in Fig. 4 (Top),we find that behavioral metrics (mean r: $0.85 \pm 0.01$) are more correlated on average than representational metrics (mean r: $0.75 \pm 0.007$), with a significant difference ($z = -7.10, p = 5 \times 10^{-8} < 0.0001$).

To further understand the relationships between different representational similarity measures, we analyzed the MDS plot (Fig. 4 (Bottom)). This visualization revealed distinct clusters of measures based on their theoretical properties. Measures that rely on inner product kernels (stimulus-by-stimulus dissimilarities) tend to group together, indicating they capture similar aspects of representational structure. On the other hand, measures that use explicit, direct mappings between individual neurons—such as Linear Predictivity and Semi-Matching—form a separate cluster. Notably, Procrustes Distance and CCA also involve alignment, similar to Linear Predictivity and Semi-Matching; however, this alignment is achieved collectively across all units or neurons rather than through independently determined mappings for each neuron. Procrustes aligns the entire configuration of points, while CCA projects the two representations onto common subspaces to maximize correlation, further distinguishing them from other representational similarity approaches.

How behavioral metrics distinguish models is crucial, as most comparative analyses of representations in neuroscience and AI revolve around understanding computations and how those computations relate to behavior; behaviorally grounded comparisons of model representations are key to this endeavor. We find that behavioral metrics distinguish between models in a consistent manner across different datasets, reinforcing the robustness of the model relationships they uncover (Appendix A.3). The

consistency of the behavioral metrics -across datasets and with each other- fulfills another scientific desiderata of replicability. Therefore, the model relationships identified by behavioral metrics are not only important but also reliable. It becomes crucial, then, to determine which representational similarity measures align with these robust behavioral relationships between models.

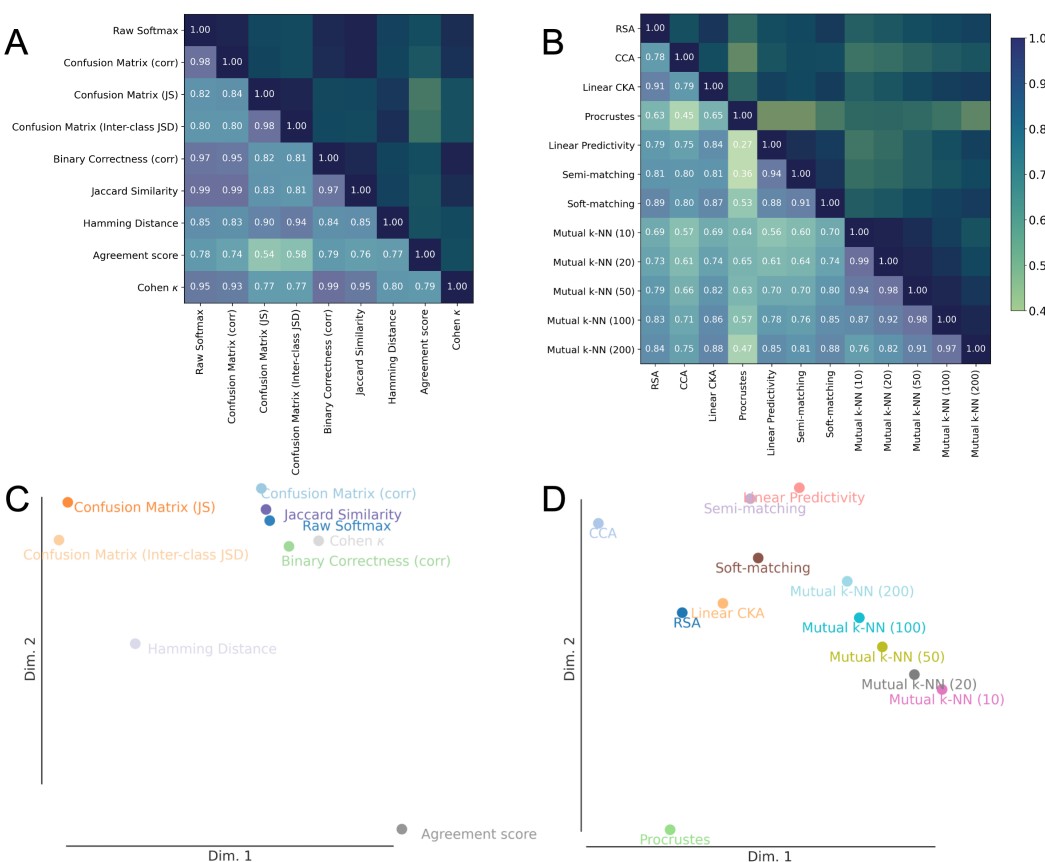

Figure 4: **Consistency Between Similarity Metrics.** (A) and (C) display the correlation matrix averaged across all behavioral datasets and the 2D-projected multidimensional scaling (MDS) plot (using 1 minus the correlation matrix as the distance matrix) for behavioral measures. (B) and (D) illustrate the average correlation matrix and the MDS plot for representational similarity measures.

## 2.4 WHICH REPRESENTATIONAL SIMILARITY MEASURES SHOW THE STRONGEST CORRESPONDENCE WITH BEHAVIORAL MEASURES?

Seeing that we want to prioritize the model relationships uncovered by behavioral metrics, we move on to investigate which –if any– representational similarity metrics reveal the same underlying relationships between models. To rigorously assess this, we computed correlations between the model-by-model similarity matrices of each representational metric with the model-by-model behavioral similarity matrix averaged across all behavioral metrics, separately for many datasets (Fig 5). We found that three metrics stood out in their alignment with behavioral metrics - RSA (mean r: $0.52$), Linear CKA (mean r: $0.64$), and Procrustes (mean r: $0.70$). Going back to our original analysis, these metrics are also able to more strongly differentiate trained and untrained models (Fig 1 Top d' measures). All these representational metrics emphasize alignment in either the overall geometry or shape of representations. Alternate measures like linear predictivity and CCA, which are commonly employed in representational comparisons in neuroscience and AI, showed significantly weaker alignment with mean correlation scores of $0.26$ and $0.19$ respectively. Given the opacity of neural representations, selecting appropriate representational similarity metrics can be challenging; these findings offer crucial guidance for metrics that support behaviorally grounded comparisons.

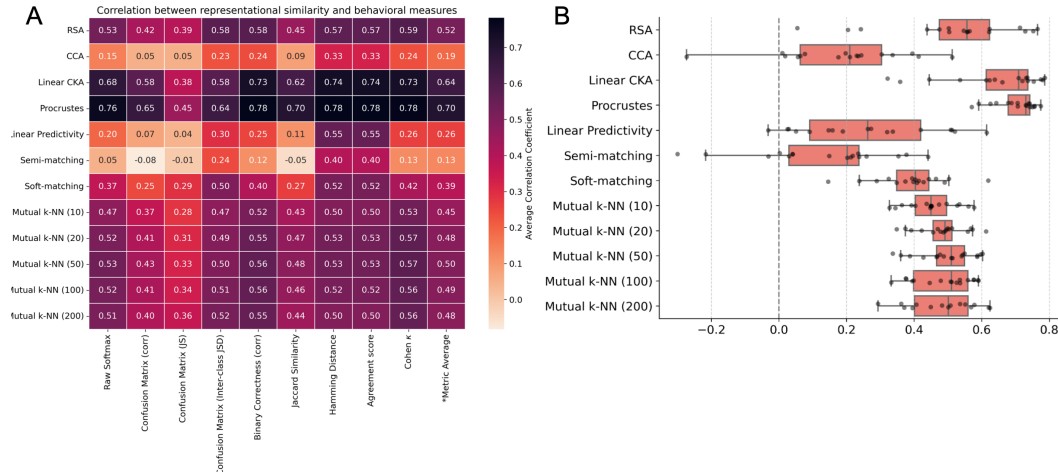

Figure 5: **Granular Comparison of Representational Similarity Measures with Behavioral Measures**: (A) Average correlation between representational and behavioral metrics across datasets. (B) Distribution of correlation scores for each representational similarity measure with behavioral measures; each point represents the averaged score for a dataset across all behavioral measures.

## 3 DISCUSSION

In this study, we compared 8 neural representational similarity metrics and 9 behavioral measures across 17 datasets. Based on the premise that behavioral differences should be mirrored in the representational structure of neural networks, we examined practical distinctions in their alignment with behavior. Metrics like RSA, CCA, and Procrustes distance, which preserve the overall geometry of neural representations, tend to align closely with behavioral measures. In contrast, methods like linear predictivity, which align dimensions without preserving global geometry, show weaker alignment. This divergence likely arises because linear predictivity has the capacity of mapping complex, distributed geometric structures to simpler, compressed ones while maintaining prediction accuracy. For instance, trained networks were observed to predict untrained network activation patterns well, yielding high symmetrized scores.

Moreover, while different behavioral measures generally show consistency, neural representational similarity metrics do not, underscoring the need for a deeper understanding of how these representational metrics discriminate between models in practical applications. Our analysis sets a new standard for representational similarity measures in neuroscience and AI, using downstream behavioral robustness as a guide for selecting the most suitable metric. This framework is especially crucial in model-brain comparisons, where representational analyses are frequently applied to assess if artificial neural networks and biological systems are serving comparable functional roles in terms of perceptual and cognitive processes.

Our framework for representational metric selection, though robust, makes some key assumptions. It assumes a specific mechanism for how behavior is 'reading out' from neural representations, and different readout mechanisms could reveal qualitatively different relationships between models. For example, applying biologically-inspired constraints, such as sparsity, could reveal divergent relationships, especially if some models encode behaviorally relevant information in a sparse manner that others do not. In such cases, the precise representation structure at the unit-level becomes critical. Additionally, we defined "behavior" within the scope of object classification across multiple out-of-distribution (OOD) image datasets. Extending evaluations to include fine-grained visual discrimination or broader tasks beyond categorization would better capture the full range of visual processing. Lastly, a stronger theoretical framework explaining why certain similarity measures align more closely with behavior than others is currently lacking in our work, but this remains an exciting direction for future research.

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

# A APPENDIX

## A.1 DOWNSTREAM BEHAVIORAL DATASETS

All datasets, directly drawn from Geirhos et al. (2019); Wang et al. (2019); Geirhos et al. (2021), share the coarser 16 labels from ImageNet. These consist of a subset of the ImageNet1k validation set sampled from the following categories: Airplane, Bear, Bicycle, Bird, Boat, Bottle, Car, Cat, Chair, Clock, Dog, Elephant, Keyboard, Knife, Oven, Truck.

- **Colour**: Served as a baseline in-distribution dataset, with half of the images randomly converted to greyscale and the rest kept in original color. Includes a total of 1280 images (80 images per label).

- **Stylized ImageNet (SIN)**: Textures from one class are applied to shapes from another while maintaining object shapes. Shape labels are used as "true labels" for confusion matrix and correctness analyses. Includes a total of 800 images

- **Sketch**: Contains cartoon-styled sketches of objects from each class, totaling 800 images.

- **Edges**: Created from the original dataset using the Canny edge extractor for edge-based representations. Includes a total of 160 images

- **Silhouette**: Black objects on a white background, generated from the original dataset. Includes a total of 160 images

- **Cue Conflict**: Images with texture conflicting with shape category, generated using iterative style transfer  (Gatys et al., 2016) between Texture dataset images (style) and Original dataset images (content). Includes a total of 1280 images.

- **Contrast**: Variants of images adjusted for contrast levels. Includes a total of 1280 images.

- **High-Pass/Low-Pass**: Images filtered to emphasize either high-frequency or low-frequency components using Gaussian filters. Includes a total of 1280 images per dataset.

- **Phase-Scrambling**: Images had phase noise added to frequencies, creating different levels of distortion from 0 to 180 degrees. Includes a total of 1120 images.

- **Power-Equalisation**: Images were processed to equalize the power spectra across the dataset by setting all amplitude spectra to their mean value. Includes a total of 1120 images.

- **False-Colour**: Images had colors inverted to their opponent colors while keeping luminance constant using the DKL color space. Includes a total of 1120 images.

- **Rotation**: Images are rotated by 0, 90, 180, or 270 degrees to test rotational invariant robustness. Includes a total of 1120 images.

- **Eidolon I, II, III**: Images distorted using the Eidolon toolbox, varying coherence and reach parameters to manipulate local and global image structures. Each filtering intensity level contains 1280 images.

- **Uniform Noise**: White uniform noise added to images with a varying range to assess robustness; pixel values exceeding bounds were clipped. Includes a total of 1280 images.

## A.2   INTER VS INTRA GROUP STATISTIC MEASURES USING $d'$ SCORES

To quantify a comparative metric's ability to reflect the expected proximity between similarly trained models, compared to their dissimilarity with the untrained models, involves speculating the group statistics from the resulting similarity matrix. We employ the $d'$ score defined as:

$$d' = \frac{\mu(A) - \mu(B)}{\sqrt{\frac{\sigma_A^2 + \sigma_B^2}{2}}}$$

where **A** represents the set of similarity scores from **intra-group** comparisons, specifically the similarity scores between every pair of trained models. **B** represents the set of similarity scores from **inter-group** comparisons, specifically the similarity scores between each pair of trained and untrained models. Equivalent to the set of entries located at the intersection of trained model rows and untrained model columns in the model-by-model similarity matrix of the metrics.

A similarity metric with $d' \geq 0$ of greater magnitude indicates a greater ability to separate trained models from untrained ones. A metric with $d' = 0$ or $d' < 0$ indicates that there were no discernible difference in average similarity scores computed in "trained model pairs" and "trained vs. untrained model pairs", or that trained vs. untrained models exhibit even higher similarity than that among trained models.

Similarly, when examining architectural differences, **A** represents intra-group comparisons within Convolutional models, while **B** captures inter-group comparisons between Convolutional models and Transformers.

## A.3   DATASET CONSISTENCY

To assess consistency across behavioral datasets, we used an $M \times M$ correlation matrix, where $M$ is the number of datasets. Each entry $i, j$ represents the correlation between datasets $i$ and $j$, derived from their downstream similarity matrices. Averaging these scores across all behavioral measures revealed high correlations, indicating consistent uniformity across most datasets.

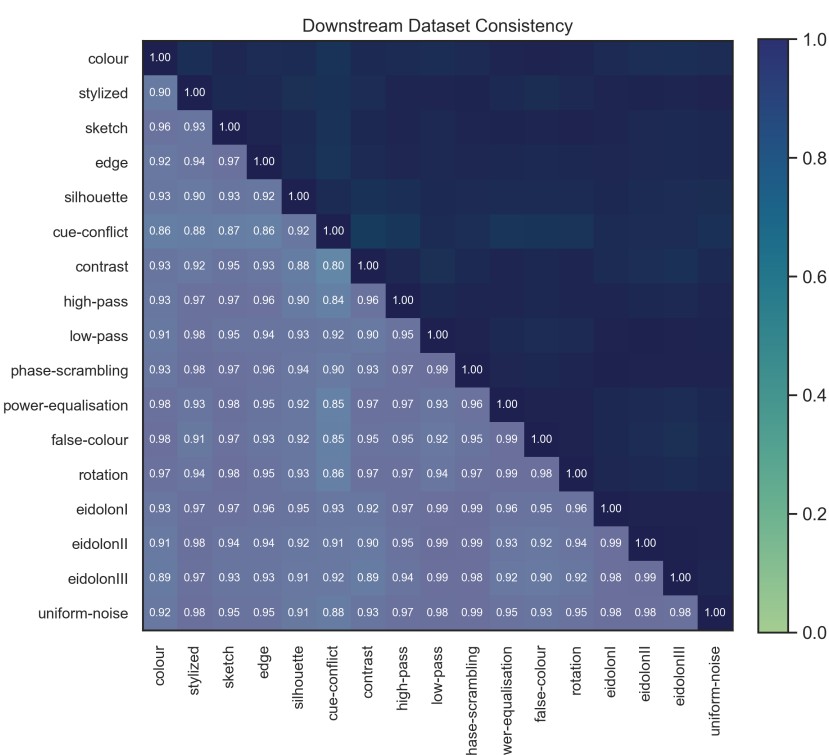

## A.4 REPRESENTATION SIMILARITY MATRICES

We include the Model-by-Model Similarity Matrix given by the 8 distinct representation measures. The scores provided are averaged across 17 datasets. For mutual k-NN, different neighborhood sizes ($k$) are included. Note that the "$1 - \text{Procrustes}$" score can range from $(-\infty, 1]$, whereas all other metrics yield scores within the range $[0, 1]$.

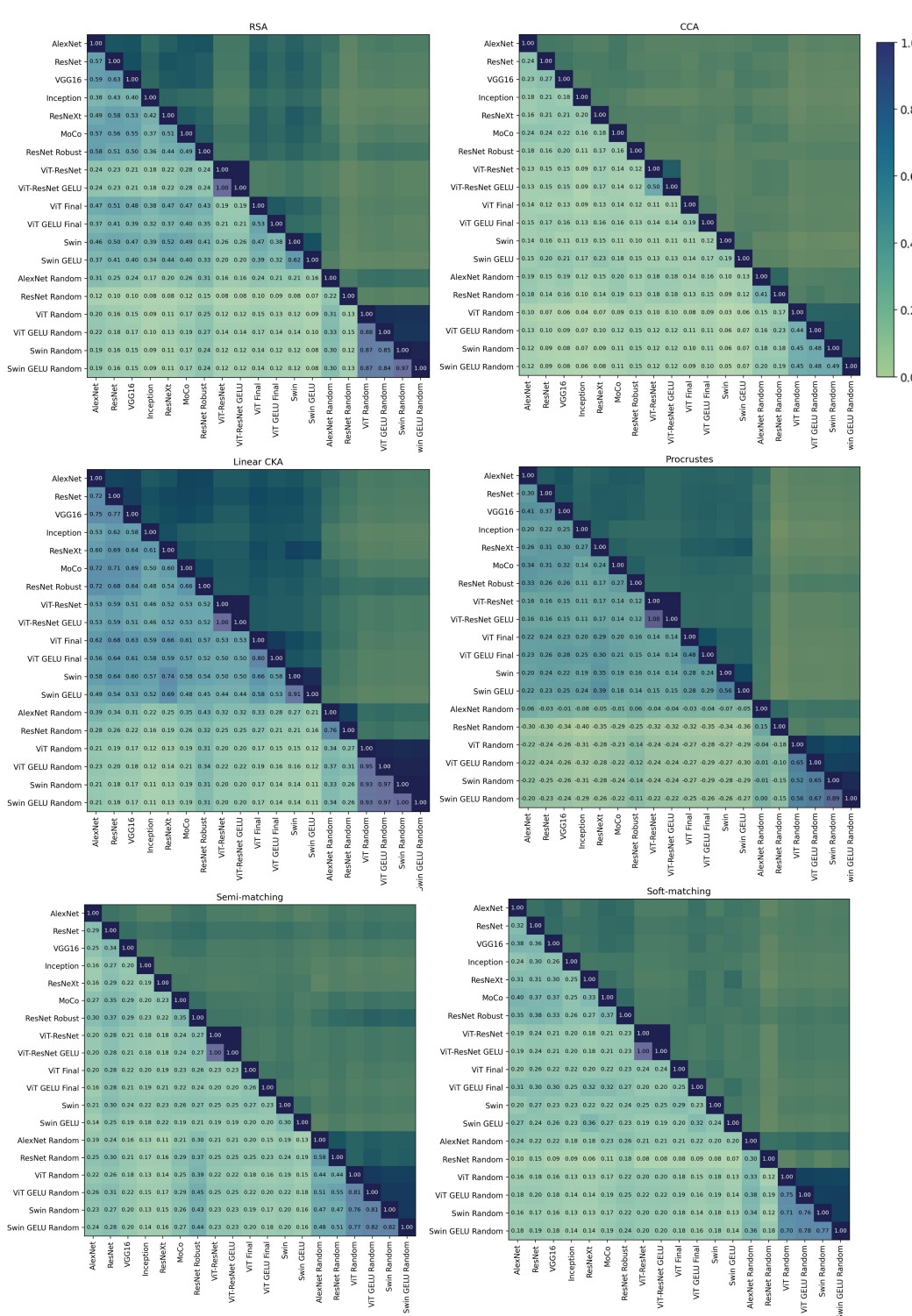

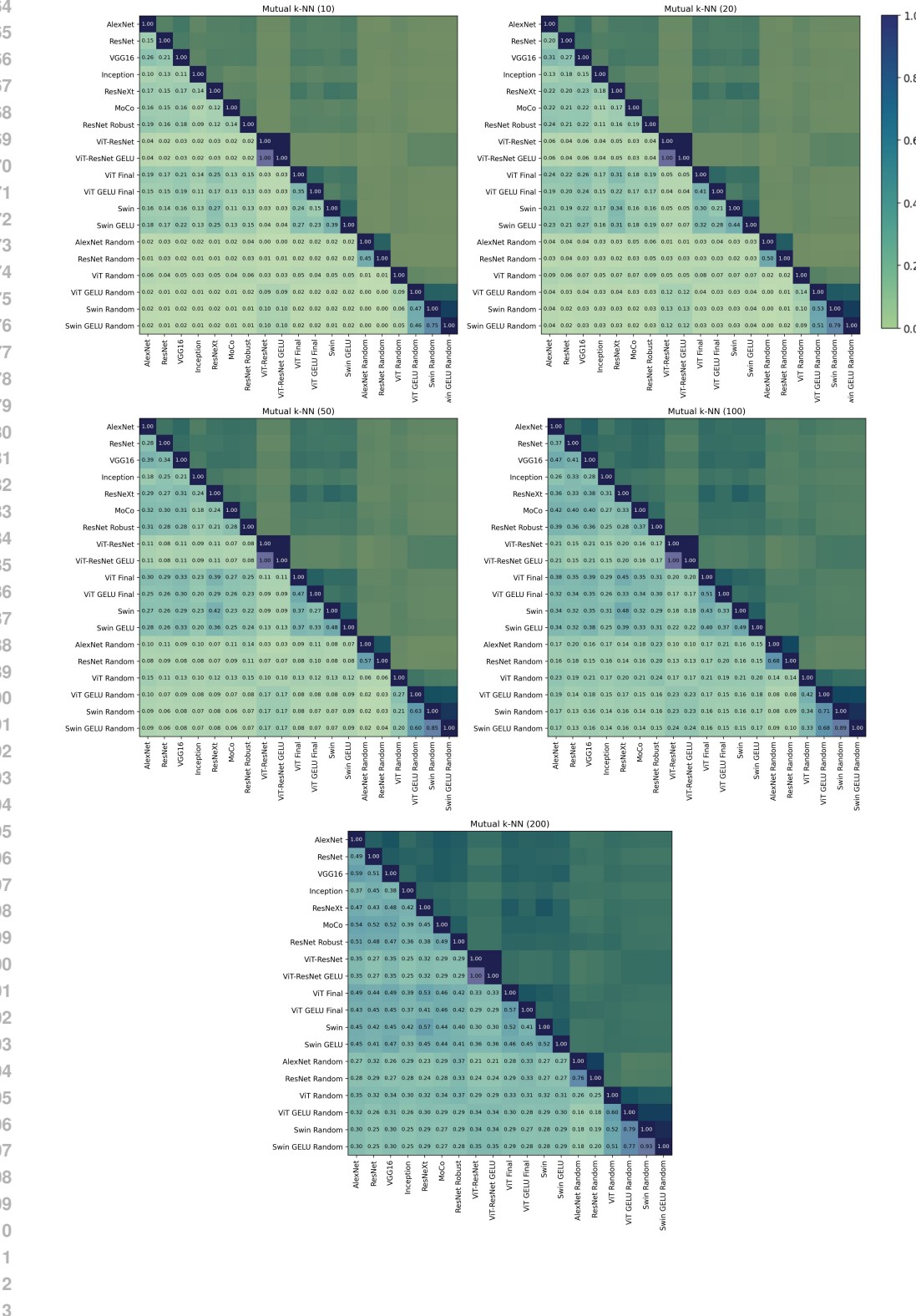

## A.5 BEHAVIORAL SIMILARITY MATRICES

Similarly, we include the Model-by-Model Similarity Matrix given by the 9 distinct behavioral measures. The scores are averaged across 17 datasets. For the measures "$1 - $ Hamming Distance" and "Agreement Scores", the alignment value can all range from $(-\infty, 1]$, whereas all other measures yield scores within the range $[0, 1]$.

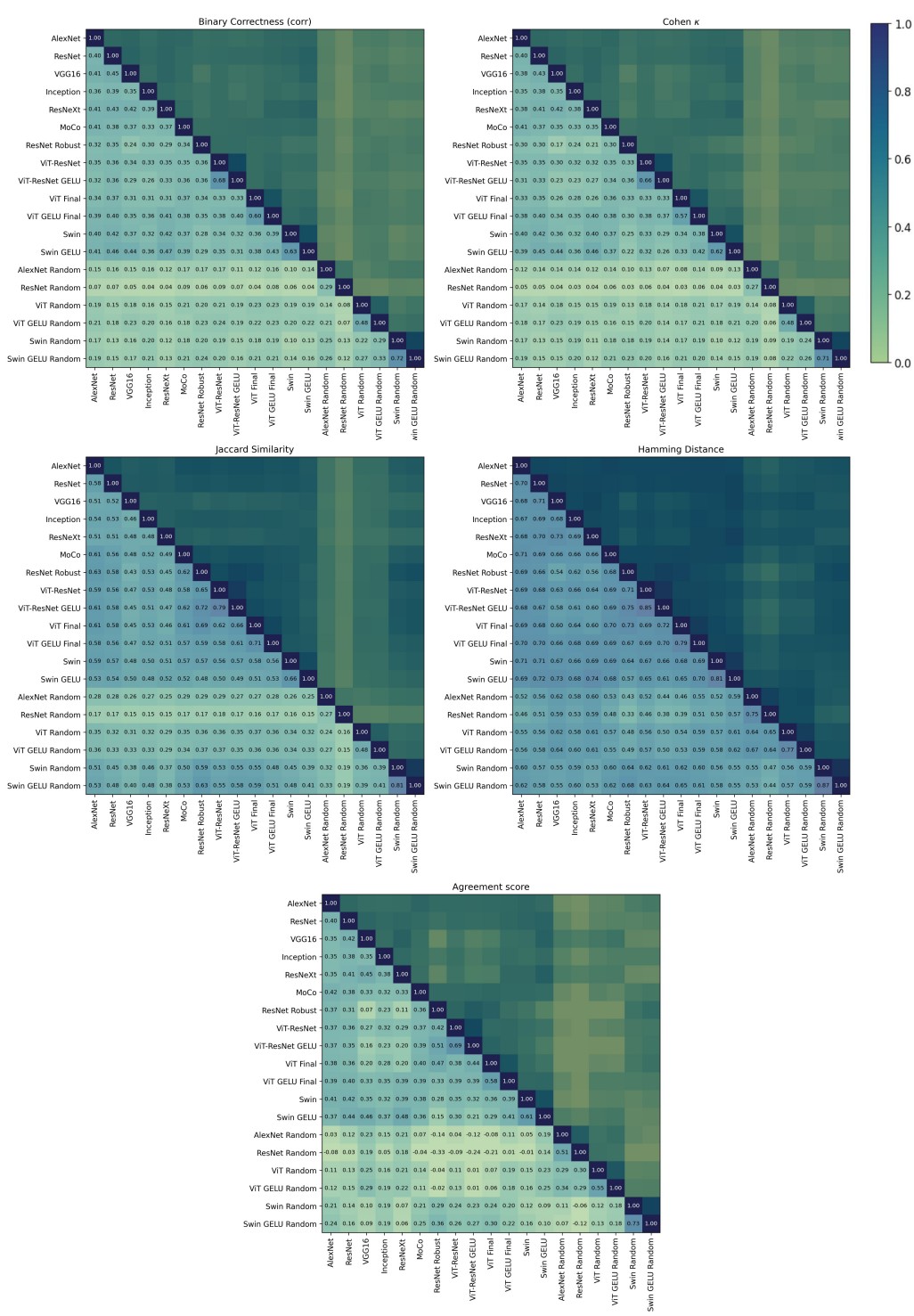

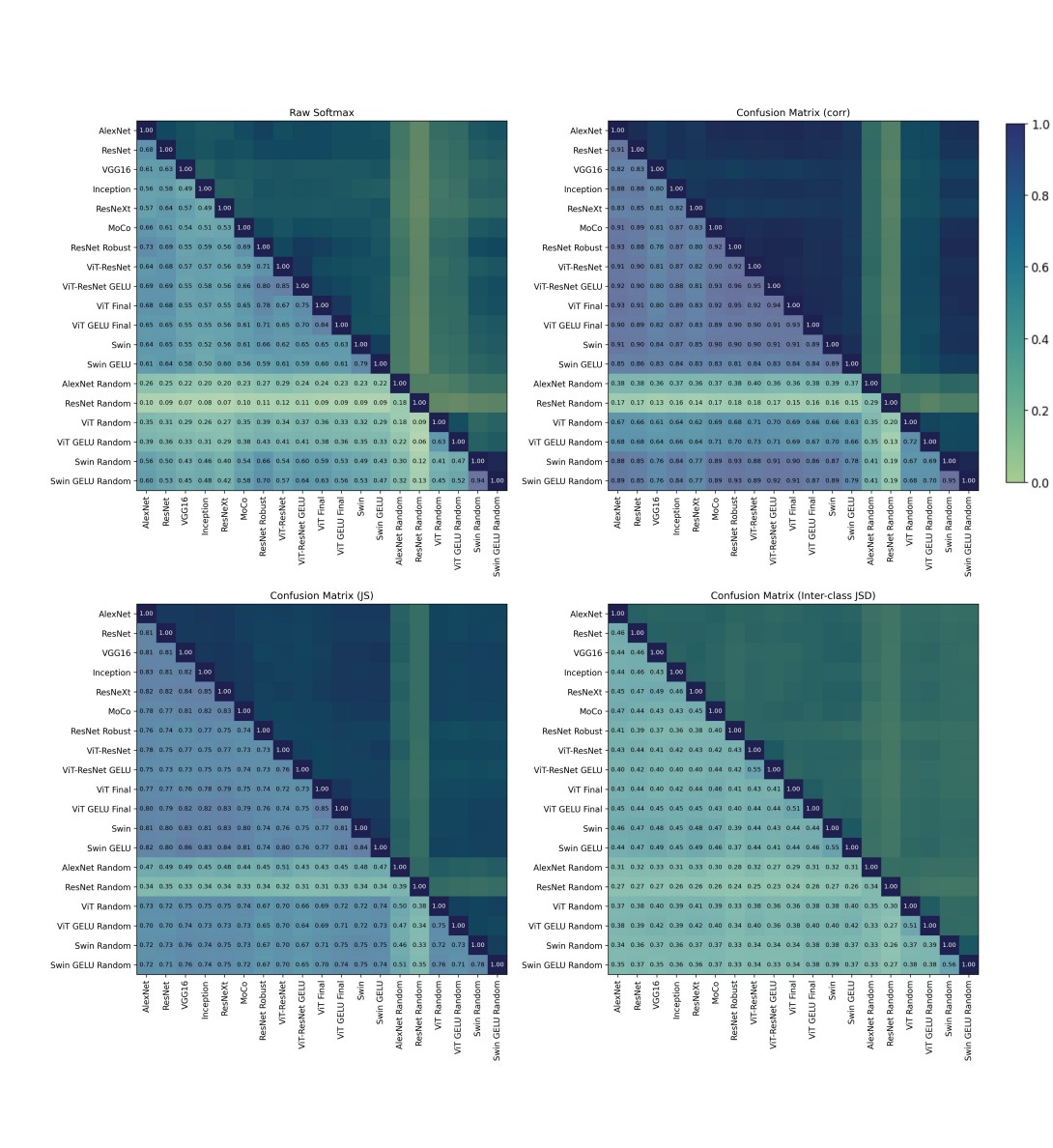

