# OpenReview forum: "Evaluating Representational Similarity Measures from the Lens of Functional Correspondence"
_ICLR.cc/2025/Conference — Submitted to ICLR 2025_

### Official Review · Reviewer_B9xm · 2024-10-31

**Soundness:** 2
**Presentation:** 2
**Contribution:** 1
**Rating:** 3
**Confidence:** 5

**Summary:**

The authors performed a systematic numerical investigation of different metrics of similarity/dissimilarity/distance between representations learned in deep neural networks.
The thesis is that understanding such metrics are important for both AI and neuroscience.
The contribution of the paper is to view this problem through the lens of ‘behavioral outcomes’. The authors conclude that two metrics (Procrutes and linear Centered Kernel Alignment), which emphasize overall geometric structure, are best aligned with behavioral outcomes.

**Strengths:**

While this paper is well written, systematic, and of potential interest,

**Weaknesses:**

there are major issues that preclude acceptance in its current form.
There is no technical novelty at all. The authors simply compare existing methods.
While the numerical investigation is systematic,
The emphasis on methods that align with network outputs (i.e., behaviors) is parochial and poorly motivated. One could easily argue that being able to distinguish between different models with similar outputs is more interesting as it distinguishes between different mechanisms that produce the same outcome. IMHO, from a neuroscience perspective, this is actually the more interesting and important desiderata. A much more balanced approach to the presentation of the results is necessary in this manuscript.

**Questions:**

none

---

### Official Review · Reviewer_RvCj · 2024-11-01

**Soundness:** 3
**Presentation:** 3
**Contribution:** 2
**Rating:** 3
**Confidence:** 2

**Summary:**

This work seeks to answer the question: what does representational alignment between two models imply about their behavioral alignment? To address this question, they apply representational similarity metrics and behavioral metrics (based on object recognition) to a collection of computational models that include both trained and untrained models. They find that linear CKA and Procrustes are more correlated with behavioral metrics, whereas methods such as linear predictivity and CCA are much less correlated with behavioral metrics.

**Strengths:**

There are several representational similarity metrics in the literature and there is relatively little understanding of their functional relationship. This work addresses this problem by comparing several metrics on a variety of behavioral tasks and models. The main strength of this paper is that it evaluates several similarity metrics (a total of 8 or 12 depending on if each k-NN is counted separately) and behavioral metrics (a total of 9). Thus, this works offers benchmarks for the representational alignment community as well as guidance for the experimental neuroscience and machine learning communities that use representational similarity metrics to evaluate representations.

**Weaknesses:**

In many ways, this paper seems incomplete. Much of the paper reads like a methods paper even though no new methods are introduced. Arguably, much of sections 1.1 and 1.2 could be relegated to the appendices. Apart from measuring the correlation between similarity metrics and behavioral metrics, there is little interpretation or investigation in the results section. In this way, this paper has the feel of exploratory analysis without follow up scientific hypothesis and analyses. Overall, I think this is a contribution that practitioners may find useful, but it could be significantly improved with scientific analysis of their findings.

**Questions:**

- Why does Procrustes correlate well with the behavioral metrics but CCA does not? My understanding is that these are part of a single family of representational metrics with the main difference being that CCA first whitenings the representation before aligning them (Williams et al. 2021). If so, what can we conclude about the model representations?
- Some of the metrics are negatively correlated with behavioral metrics! What's going on there? Is this just noise?

---

### Official Review · Reviewer_ipFd · 2024-11-01

**Soundness:** 2
**Presentation:** 3
**Contribution:** 2
**Rating:** 3
**Confidence:** 4

**Summary:**

The authors used a collection of representational similarity measures to the activations in the penultimate layers of a large collection of deep neural networks trained on ImageNet. They then compare the results they get align with the results based on comparisons of the outputs of the network, i.e. their behaviour. They observe that the behavioural measures discriminate untrained vs trained networks well, but struggle to distinguish architectures, are are more related to RSA, linear CKA and Procrustes alignment than to nearest neighbour analyses and linear predictions and CCA measures.

**Strengths:**

Deeper understanding of the representational similarity measures is an important topic.

 Comparisons to the actual classification behaviour of the models is an interesting new viewpoint.

The basic results fit with earlier analyses and appear to be solid.

**Weaknesses:**

While I generally believe the results and they are somewhat interesting, I think the analyses could be much more thorough and broad.

The formula for RSA comparisons is wrong. To represent the classic formulation of RSA requires a standardisation of the X values and an important vectorisation step implied in $\tau$. And there are many more modern and preferred variations of this technique today. Similarly the linear encoding model description given here says nothing about the important steps of dimensionality reduction, regularisation and cross-validation. Thus, unfortunately, I have doubts whether the authors implemented a sensible state of the art version of these comparisons. Also it is clear that they are missing quite a few methods that are used in practice.

The authors focus only on penultimate layers in imagine trained networks. This is a small subset of the representations we are interested in studying with representational similarity measures. Especially, how the relation to behavioural measures brakes down when earlier layers are used would be very informative.

As the authors admit, they do not provide a theoretical understanding why some measures are more or less aligned with behaviour. There is some literature discussing connections and differences between the different similarity measures. Thus, there would be something to discuss for sure.

Most of the paper covers corse overall patterns in the data. I would have appreciated a deeper dive into what networks (pairs) the measures disagree about, on which tasks they behave differently, etc. Perhaps even some theoretical insight could be found there.

**Questions:**

I found the high correlations in A.3 quite surprising. How does it come that all tasks produce so similar similarity structure among models?

---

### Official Review · Reviewer_BLDW · 2024-11-07

**Soundness:** 3
**Presentation:** 3
**Contribution:** 2
**Rating:** 5
**Confidence:** 4

**Summary:**

The authors assess the degree of correspondence between various representational similarity measures and task metrics for networks.  They show that certain representational metrics (CKA and Procrustes) distinguish trained from untrained models and align with task metrics.  Other representational metrics (linear predictivity) did not align well with task metrics.  In total they examine 19 models, 17 datasets, 9 task measures, and 8 representational measures.

**Strengths:**

The paper is mostly straightforward and clear.  Although an empirical study, it tackles a useful question.  Figures 4 and 5 provide a clear and simple message.

**Weaknesses:**

The captions for each figure can be improved by including more information about what properties are being averaged over and what are being correlated (e.g. the dots in Figures 3 represent datasets).

The use of the word “behavior” is awkward to this reader, particularly when (unless I’m mistaken) the only thing you are considering is classification.  This also makes the results sound much more general than they likely are.  It is certainly reasonable that the success of CKA and Procrustes in this case is due at least in part to the fact that the task in question is a discrete classification task.  It’s easy to imagine that other tasks might yield better correspondence with different representational metrics.

One has to wonder about the variability of different model instances.  Unless I’m mistaken, you’ve taken one pre-trained instance of each model.

There isn’t really any attempt at providing any theoretical justification for the results.  I realize that’s a tall order and don’t see it as necessary here, but it would be nice to have some understanding of why Procrustes and Linear CKA show such good agreement.

**Questions:**

What can you say about the variability of these metrics over different model instances?

Can you apply this to other types of tasks beyond classification?  Do they still show good agreement with Procrustes and Linear CKA?

---

### Meta-Review · Area_Chair_CD71 · 2024-12-21

**Metareview:**

This paper examines the alignment between representational similarity metrics used for assessing similarity of neural and behavioral datasets.  While the reviewers agreed that it was clearly written and addresses a useful question, they unfortunately raised significant concerns about novelty, thoroughness and the lack of theoretical interpretation or insight. I regret that the paper in its current form cannot be accepted to this year's ICLR, but I wish the authors the best of luck in revising it for publication elsewhere.

**Additional Comments On Reviewer Discussion:**

The authors did not write a rebuttal, so there was no discussion during the rebuttal period.

---

### Decision · Program_Chairs · 2025-01-22

Reject